# Joint Intra/Inter-Slot Code Design for Unsourced Multiple Access in 6G Internet of Things

**DOI:** 10.3390/s23010242

**Published:** 2022-12-26

**Authors:** Yuanjie Li, Kai Niu, Chao Dong, Shiqiang Suo, Jiaru Lin

**Affiliations:** 1Key Laboratory of Universal Wireless Communications, Beijing University of Posts and Telecommunications, Beijing 100876, China; 2CICT Mobile Communication Technology Co., Ltd., Beijing 100190, China

**Keywords:** unsourced multiple access, T-Fold IRSA, IDMA, degree distribution optimization, internet-of-things, 6G

## Abstract

Unsourced multiple access (UMA) is the technology for massive, low-power, and uncoordinated Internet-of-Things in the 6G wireless system, improving connectivity and energy efficiency on guaranteed reliability. The multi-user coding scheme design is a critical problem for UMA. This paper proposes a UMA coding scheme based on the T-Fold IRSA (irregular repetition slotted Aloha) paradigm by using joint Intra/inter-slot code design and optimization. Our scheme adopts interleave-division multiple access (IDMA) to enhance the intra-slot coding gain and the low-complexity joint intra/inter-slot SIC (successive interference cancellation) decoder structure to recover multi-user payloads. Based on the error event decomposition and density evolution analysis, we build a joint intra/inter-slot coding parameter optimization algorithm to minimize the SNR (signal-to-noise ratio) requirement at an expected system packet loss rate. Numerical results indicate that the proposed scheme achieves energy efficiency gain by balancing the intra/inter-slot coding gain while maintaining relatively low implementation complexity.

## 1. Introduction

### 1.1. Background

Recently, academics and industry have proposed many prospects for the evolution of the IoT (Internet-of-Things) in the next generation [1]. In general, the 6G IoT system mainly faces the following three challenges: The rapid growth of connectivity [2], the guarantee of low latency under specific reliability [3], and the requirement of low power consumption and low implementation complexity [4].

Based on this, Polyanskiy proposed the concept of unsourced multiple access (UMA) in 2017 [5]. The UMA removes the coordination center in the network to reduce the transmission cost and latency of the frequent access of vast short packages, which is the major drawback of traditional schemes OMA (Orthogonal Multiple Access) [6], coordinated NOMA (Non-Orthogonal Multiple Access) [7], and grant-free access [8]. The uplink channel is always available to users, and the network works in the unsourced style, adapting to large-scale frequent connection requests. As a result, the system optimization criteria need to be changed from sum-rate to PUPE (Per-User Probability of Error), that is, to achieve the massive access ability under a specific average packet loss rate. The UMA’s achievability bound was given in [5], laying the foundation for research in this field.

### 1.2. Related Works

After that, many works emerged, exploring how to approach the performance bound through practical coding design under UMA [9,10]. These schemes are based on three basic paradigms: random spreading, T-Fold Aloha [11], and T-Fold IRSA (Irregular Repeated Slotted Aloha) [12].

The random spreading scheme regards the entire transmission frame as the available length of code chips for each user. The users’ data packets are superimposed after the symbol-level spreading. Representative works are the sparse IDMA (Interleave-Division Multiple Access) [13] based on LDPC (Low-Density Parity Check Code) and Polar-RS (Random Spreading) [14] and Polar-SS (Sparse Spreading) [15] based on polar codes. As such schemes occupy the entire frame to form a low-rate code, thereby obtaining the highest coding gain, which has a similar performance to that close to the boundary. However, at the same time, their high implementation complexity is also a problem that cannot be ignored.

T-Fold Aloha divides the transmission frame into slots with equal lengths. On each slot, a multi-access code is designed to control the error rate under multi-user superposition, ensuring a decodable threshold up to *T*. A classical realization is the concatenated code scheme [16], where the outer code guarantees the T-Fold feature, and the inner code treats noise. The compressed sensing (CS) encoder in CCS (Coded Compressed Sensing) [17] and SPARC (SParse Regression Code) [18] schemes plays the above two roles at the same time and has higher coding efficiency. However, the tree code also suffers an error propagation effect, and the CS coding gain is still limited in the region of high users.

T-Fold IRSA paradigm originated from the research on contention resolution Aloha [19] by G. Liva. Its design consists of two main aspects: the T-Fold intra-slot code and the packet-level inter-slot code. Cheng et al. proposed the SC-LDPC+IRSA scheme [20], spreading the SC-LDPC (Spatially Coupled-LDPC) packets by IRSA. Polar-IRSA [21] changed the SC-LDPC to the SCL decoded multi-user polar code with better performance under the short code length and achieved higher performance gain. In addition, [22] also gives a theoretical analysis of the IRSA framework based on the finite block length bound [23] and asymptotic analysis. Such schemes have performance advantages over T-Fold Aloha; under certain conditions, their performance is close to the high-complexity random spreading schemes. However, these schemes are based on the idealized asymptotic assumptions, the exploitation of intra-slot coding gain is limited, and there is still some room compared with the achievability bound.

In addition, some works also discuss the theoretical analysis [24] and coding scheme [25] design transferred from the AWGN channel to the Rayleigh block fading channel. Besides, MIMO (Multi-Input Multi-Output) transmission [26], channel estimation, activity detection [27], and their co-design [28] push the horizon from theory to practical deployment.

### 1.3. Contributions

Following the works under the T-Fold IRSA paradigm, we continue to investigate the trade-off between the intra-slot coding gain and the inter-slot diversity gain. Besides, with the premise of realization, we attempt to obtain improvement or balance in terms of performance and complexity. The main contributions of the proposed scheme are the following aspects:**Enhanced intra-slot coding structure.** We apply the IDMA scheme with CS header to intra-slot code design. The user payload is split into two parts, encoded by IDMA and CS encoders separately, and combined as the intra-slot codeword. The CS part carries the user-specific interleaver pattern of the IDMA codeword, and the IDMA part is the multi-access code resolving the superposition interference.**Joint intra/inter-slot iterative decoder.** Under the IRSA paradigm, the superimposed spreading pattern of intra-slot code packets is expanded as a compound inter-slot factor graph, recovered by combining the CS pilot decoding results among the slots. The ESE + BP decoder of the intra-slot IDMA code is the embedded operation on the slot nodes. The inter-slot SIC iteration is performed on the graph to eliminate the interference on the slot nodes, making the overload slot nodes decodable.**Joint intra/inter-slot coding parameter optimization.** To minimize the required SNR under the given PUPE standard and a limited number of channel resources, we follow the idea of error event decomposition to build the framework of parameter optimization. The error caused by each coding module is modeled as a function of its coding parameter. Especially, the inter-slot degree distribution is analyzed by density evolution with finite-length realization and energy cost conditions. Then we integrate these error functions in a global optimization problem and design a heuristic bootstrap search algorithm to jointly optimize all these related parameters, including the intra-slot CS pilot length, the IDMA coding rate, and the inter-slot degree distribution.

### 1.4. Content Organization

This article organizes its content in the following way. Section 2 gives some basic concepts and universal notations to help clarify the whole framework. Section 3 focuses on the encoder’s design and the proposed scheme’s decoding algorithm. Next, Section 4 discusses the coding parameter optimization problem based on error analysis by decomposition and the unified joint optimization algorithm. Finally, by numerical simulation, Section 5 evaluates its energy efficiency and computational complexity under the optimized configuration.

## 2. Definitions and Notations

The core problem of UMA is to construct a coding scheme that allows $K_{a}$ unsourced active users to transmit length-*B* data payload for each on a fixed length-$N_{tot}$ frame at a target PUPE and given SNR level. By definition [5], the PUPE can be expressed as:(1)$$\epsilon =\sum_{k=1}^{K_{a}}P\{{\hat{w}}_{k}\neq w_{k}\},$$
where *k* is the index of users and ${\hat{w}}_{k}$ is the decoded version of transmitted payload w. When ϵ, $N_{tot}$, and *B* are given, for each $K_{a}$, there exists an optimal configuration for a coding scheme that makes the required SNR minimum, which is referred to as the SNR threshold. Thus, the energy efficiency of any UMA scheme can be characterized by the $K_{a}$-SNR threshold curve.

Under the T-Fold IRSA paradigm, the entire frame of length $N_{tot}$ is sliced into *V* slots with length $N=N_{tot}/V$ for each. The degree distribution λ(x) determines the distribution of the repetition rate of the intra-slot coded packets for each user. Since the packets are randomly distributed on slots, the number of superimposed packets $L_{v}$ on slot *v* is also randomized. The multi-user interference increases with $L_{v}$. T-Fold means that the intra-slot code guarantees that at most $T_{th}$ users can be correctly decoded. Thus, $T_{th}$ is the threshold for the T-Fold IRSA system.

Here are some notations for matrices and vectors. The upper-bold case A represents the matrix, and $A_{i,j}$ represents its element at the *i*-th row and the *j*-th column. The lower-bold case c represents the vector, and $v_{i}$ is the *i*-th element. The vectors are column vectors in default.

## 3. Joint Intra/Inter-Slot Coding Scheme

The structure of the proposed coding scheme is given in this section. In general, it is an intra/inter-slot nested structure, as Figure 1 shows.

The payload of each user $w_{k}$ is encoded by the intra-slot code to produce $v_{k}$, and then randomly repeated $\beta_{k}$ times to form an inter-slot packet-level spreading pattern. On the receiver side, the slot-by-slot intra-slot decoding is integrated into the packet-level SIC iteration process on the inter-slot factor graph. The intra-slot codeword $v_{k}$ contains the IDMA codeword $x_{k}$ with a CS pilot $s_{k}$ indicating both the intra-slot interleaving pattern and the inter-slot spreading pattern. The inter-slot code is an IRSA structure enabling the SIC. Thus, the proposed scheme is introduced under the above framework.

### 3.1. Encoder

The intra-slot encoder is concatenated with the inter-slot encoder. The overall scheme of the encoder is depicted in Figure 2a. For each user *k*, the intra-slot coded packets are the non-zero ’chips’ of the inter-slot code. The intra-slot codeword consists of the CS pilot and the IDMA-coded part, each carrying part of the payload. Due to the unsourced feature, users must adopt a common codebook. Therefore, the CS pilot not only determines the user-specific configuration of the intra-slot code but also represents the structure of the inter-slot code, which are both randomized to deal with the multi-user interference.

#### 3.1.1. Intra-Slot Encoding

The permutation pattern of each user should be known at the receiver side to enable the decoding process, by transmitting it as an encoded pilot attached ahead of the IDMA data coded part. Thus, we slice $w_{l}$, the information payload of user *l*, into two parts: the pilot info sequence $d_{l}=\left[w_{l,1},\cdots ,w_{l,B_{s}}\right]$ of length $B_{s}$ that carries both the permutation pattern and part of the user info, along with the data info sequence $b_{l}=\left[w_{l,B_{s}+1},\cdots ,w_{l,B}\right]$ of length $B_{c}=B-B_{s}$ for IDMA encoder.

The function of the CS encoder is to map $d_{l}$ into pilot $s_{l}$. The binary sequence $d_{l}$ is firstly converted to decimal $\tau_{l}$. Then $\tau_{l}$ is through a bijective map to the column index of $M_{s}$-by-$N_{s}$ sensing matrix A, usually configured as a normalized Gaussian random matrix. As the length of A is $B_{s}$, the number of columns of A, $N_{s}\geq 2^{B_{s}}$. A natural but effective mapping is to choose the $\tau_{l}+1$-th column of A as the length-$N_{s}$ pilot, i.e., $s_{l}=a_{\tau_{l}+1}$.

The data sequence $b_{l}$ is sent to a common rate-$R_{L}$ LDPC encoder identical among all users, generating codeword bit sequence $u_{l}$ with length $B_{u}=B_{c}/R_{L}$. To introduce intra-slot coding diversity, $u_{l}$ is repeated $R_{r}$ times to produce a low rate codeword $c_{l}$ with length $B_{c}$, where the repetition rate is also the same for all users. The next step is the user-specific interleaver. As mentioned before, the permutation pattern is determined by $d_{l}$. The decimal $\tau_{l}$ converted from $d_{l}$ is used to choose the τ-th pattern $f_{\tau_{l}}\in F$ in the common pattern set F. Through permutation function $c_{l}^{\prime }=f_{\tau_{l}}\left(c_{l}\right)$, we get the interleaved bit sequence $c_{l}^{\prime }$. According to constellation G, $c_{l}^{\prime }$ is modulated to IDMA codeword symbol sequence $x_{l}$ with length $N_{c}=B_{c}/\left(R_{L}*R_{r}*N_{m}\right)$, where the size of the constellation is $Q_{m}=\left|G\right|$, and the modulation order is $N_{m}=\log_{2}\left(Q_{m}\right)$. Especially, $N_{c}=B_{c}/\left(R_{L}*R_{r}\right)$ for BPSK (Binary Phase Shift Keying).

Finally, $x_{l}$ and $s_{l}$ are stitched together to form the whole packet of user *l*, i.e., $v_{l}=\left[b_{l};d_{l}\right]$. After assembling, the length of $v_{l}$ is $N=N_{s}+N_{c}$. In general, the intra-slot encoder is common for every user. This not only keeps the simplicity of the encoding process but also ensures the common codebook requirement of unsourced settings. The ability of intra-slot code to distinguish superimposed user packets mainly comes from the compressed sensing pilot code and IDMA’s user-specific random interleaver, which are all dependent on the randomness of the pilot info slice $d_{l}$. The whole structure of the intra-slot encoder is shown in Figure 2b.

#### 3.1.2. Inter-Slot Irregular Spreading

The inter-slot encoding process is based on the intra-slot code, which adds irregular spreading diversity among different user packets by random scheduling, while the encoder remains the same structure shared by all users. Each user *k* determines the number of packet repeats $\beta_{k}$ based on the local random scheduler, making the distribution of beta $P(\beta_{k}=i)$ approaches the preset λ(x). Under the IRSA scheme, λ(x) is the distribution polynomial of $\beta_{k}$, written as:(2)$$\lambda \left(x\right)=\sum_{i=1}^{I_{\max}}\lambda_{i}x^{i}=\sum_{i=1}^{I_{\max}}P\left(\beta_{k}=i\right)x^{i},$$
where $\lambda_{i}$ is the probability of user node of degree-*i* on the packet superposition factor graph and satisfies the normalized constraint $\lambda \left(1\right)=\sum_{l=1}^{I_{\max}}\lambda_{i}=1$. Therefore, the random scheduler generates $\beta_{k}∼\lambda \left(x\right)$ and then creates a vector:(3)$$\delta_{l}^{\prime }=\left[\underset{\beta_{k}}{\underset{︸}{1,1,\cdots ,1}},\underset{V-\beta_{k}}{\underset{︸}{0,0,\cdots ,0}}\right],$$
which is then randomly permuted to ensure the 1-elements uniform, producing the $\beta_{k}$-sparse irregular diversity mapping vector $\delta_{k}$. Let the intra-slot codeword $v_{k}$ take Kronecker product with spreading pattern $\delta_{k}$:(4)$$z_{k}=v_{k}⊗\delta_{k}.$$

$v_{k}$ is sent to the slot where the element is one in $\delta_{k}$. When β=1 the packet is not repeated, and when β>1 the packet gains repetition diversity. Each user follows the above structure and superimposes their spread packets $z_{k}$ in the AWGN (Additive White Gaussian Noise) channel. The received signal $\bar{y}$ is:(5)$$\bar{y}=\sum_{k=1}^{K_{a}}z_{k}+n,$$
where n is the Gaussian noise with variance $\sigma^{2}$. Through this inter-slot encoding, the repeated user packets on different slots form a sparse spreading structure, providing packet-level diversity gain.

### 3.2. Joint Intra/Inter-Slot Decoder on Compound Factor Graph

Accordingly, the receiver follows the mirrored structure. The inter/intra-slot coding configurations are first recovered by pilot decoding, and then the intra-slot decoder is embedded as an operation on the slot nodes of the inter-slot SIC procedure. The receiver will be introduced in the following subsections under the above framework.

#### 3.2.1. Intra-Slot Decoder

The iteration begins with the intra-slot packet decoding process. The received packet on each slot *v* can be sliced from the whole received signal $\bar{y}$:(6)$$y_{v}=\left[{\bar{y}}_{(v-1)N+1},\cdots ,{\bar{y}}_{vN}\right]=\sum_{l=1}^{L_{v}}\gamma_{v,l}+n,$$
where $L_{v}$ is the number of superimposed user on slot *v*. According to the intra-slot encoding structure, $y_{v}$ can be further separated into two parts: the CS header $y_{v}^{s}=\left[\gamma_{v,1},\cdots ,\gamma_{v,N_{s}}\right]$ and the IDMA signal $y_{v}^{c}=\left[\gamma_{v,N_{s}+1},\cdots ,\gamma_{v,N}\right]$. Since the IDMA decoding requires the interleaver pattern, the CS pilot is decoded first. $y_{v}^{s}$ can be expanded by:(7)$$y_{v}^{s}=\sum_{l=1}^{L_{v}}s_{v,l}+n=\sum_{l=1}^{L_{v}}a_{\tau_{l}+1}+n=\sum_{l=1}^{L_{v}}Ae_{\tau_{l}+1}+n=Ag_{v}+n,$$
where $e_{\tau_{l}+1}$ is an 1-sparse vector with all-zeros elements except for position $\tau_{l}+1$, and $g_{v}=\sum_{l=1}^{L_{v}}e_{\tau_{l}+1}$ is an *L*-sparse vector, assuming no resource collision among users. By sending $y_{v}^{s}$ to the support recovery algorithm [29], the column index set of A, as we modeled, the support set ${\hat{D}}^{s}$, is searched out. Remap the elements in ${\hat{D}}^{s}$ back to decimals ${\hat{\tau }}_{l}$ by aligning the ${\hat{\tau }}_{l}-1$-th column to ${\hat{\tau }}_{l}$. Then convert the decimals ${\hat{\tau }}_{l}$ to length-$B_{s}$ binary sequences ${\hat{d}}_{l}$, which are the pilot info parts of user payloads. Besides, the interleaver patterns $f_{{\hat{\tau }}_{l}}$ are recovered by selecting the τ-th pattern $f_{{\hat{\tau }}_{l}}\in F$.

After the recovery of $f_{{\hat{\tau }}_{l}}$, the IDMA decoding can be started. Since the intra-slot repetition rates are the same among users, we can utilize a simple linear algorithm, the ESE + BP iterative structure [30], as depicted in Figure 3.

The ESE + BP iteration may continue for a specific number of times $I_{\max}^{1}$, and then the final hard-decision output ${\hat{b}}_{l}$ at the BP decoder of each branch is stitched together with the corresponding pilot info part ${\hat{d}}_{l}$, reconstructing the complete payload decoding result ${\hat{w}}_{l}$.

#### 3.2.2. Inter-Slot Decoder

To recover the superimposed multi-user packets on the overload slots, the inter-slot decoding performs SIC on the compound packet-level factor graph known by the reconstruction process. After the initial intra-slot decoding on all slots, the CS pilots should be recovered as pilot info ${\hat{d}}_{v}$, while the corresponding IDMA parts ${\hat{b}}_{v}$ are not all successfully decoded, especially for those on the overload slots.

Using the CS pilots as pointers, the repetition relationship can be confirmed by comparing ${\hat{d}}_{v,k}$ in one slot with ${\hat{d}}_{v_{1},k_{1}}$ on the others. According to Section 3.1.2, the number of unique pilots among all slots is the number of users $K_{a}$. For user packet *k*, the indexes of the slots where a replica of it exists form a set:(8)$$U_{k}=\left\{v_{1}|{\hat{d}}_{v,k}={\hat{d}}_{v_{1},k},v\neq v_{1}\right\},$$
where $\max|U_{v}|=I_{\max}$. We combine those sets $U_{1},\cdots ,U_{K_{a}}$ as adjacent matrix U, where the edge between user node *k* and slot node *v* is determined. Thus, the intra-slot packet spreading structure can be recovered as the compound factor graph in Figure 4. And The process of the inter-slot decoding described in Algorithm 1 is performed on this graph structure.
**Algorithm 1** Inter-slot SIC decoding on the compound packet-level factor graph**Require:** The received signal $y_{v}$ on each slot *v*, compressed sensing matrix A, interleaver set F, consteallation A, noise variance $\sigma^{2}$, threshold $T_{th}$, maximum SIC iteration *J*.**Ensure:**  Decoded payloads of all users ${\hat{w}}_{k}$.1:Initialization: Perform the intra-slot decoding in Section 3.2.1 on each slot, and get the pilot info ${\hat{d}}_{v,k}$, the IDMA decoding result ${\hat{b}}_{v,k}$, and the superimposed number of users $L_{v}^{1}=\left|{\hat{D}}_{v}^{s}\right|$ on each slot;2:Factor graph reconstruction: Compare all the CS pilots ${\hat{d}}_{v,k}$, and combine the repetition relationship sets $U_{l}$ into adjacent matrix U by (Equation 8);3:**while** 
j≤J 
**do**4:   Update the slot nodes subsets $V_{j}^{+}$ and $V_{j}^{-}$ by (Equation 9), and the user node subsets $K_{j}^{+}$, $K_{j}^{-}$ and $K_{v^{\prime }}^{+}$ accordingly;5:   **for all** $v^{\prime }\in V^{-}$ **do**6:     **if** $L_{v^{\prime }}^{j}-K_{v^{\prime }}^{+}\leq T_{th}$ **then**7:       Forward message Passing (from user node *l* to slot node $v^{\prime }$);8:       Remap the user massages in the effective edge set ${\hat{b}}_{k},k\in K_{v^{\prime }}^{+}$ to IDMA packets $x_{k}^{\mathrm{SIC}}$;9:       Peel off the known interference $x_{k}^{\mathrm{SIC}}$ as (Equation 11);10:      Backward Message Passing (from slot node $v^{\prime }$ to user node *k*);11:      Intra-slot decoding: Perform the IDMA decoding part on slot $v^{\prime }$, get the recovered user information ${\hat{b}}_{v^{\prime },k}$;12:      User node update: Add/Subtract the newly recovered user on slot $v^{\prime }$ in $K_{j+1}^{+}$/ $K_{j+1}^{-}$;13:      Update the slot counter $L_{v^{\prime }}^{j+1}=L_{v^{\prime }}^{j}-\left|K_{v^{\prime }}^{+}\right|$.14:    **else**15:      SIC not startded, reserve the slot counter $L_{v^{\prime }}^{j+1}=L_{v^{\prime }}^{j}$.16:    **end if**17:  **end for**18:**end while**

At the *j*-th SIC iteration, according to the T-Fold IRSA model, the slot nodes set can be separated into two subsets by the given threshold $T_{th}$
(9)$$\begin{array}{c} v\in \left\{v|L_{v}^{j}\leq T_{th}\right\}=V^{+}\\ v^{\prime }\in \left\{v^{\prime }|L_{v^{\prime }}^{j}>T_{th}\right\}=V^{-} \end{array}$$
where the initial slot node degree $L_{v}^{1}=\left|{\hat{D}}_{v}^{s}\right|$. Also, the user nodes set has two subsets, the successfully decoded set $K_{j}^{+}$ and the undecoded set $K_{j}^{-}$. For each underload slot node $v\in V^{+}$ that satisfies the decodable condition, add its adjacent user node *k* into $K_{j}^{+}$. Based on this graph, the message passed from the user node to the slot node is the remapped decoded IDMA packet $x_{k}^{\mathrm{SIC}}$. After the SIC peeling session, if the degree of an overload slot node can be reduced under the threshold $T_{th}$, this slot will be decodable, where its output message is the reliable decoded IDMA info parts ${\hat{w}}_{k}$. Define the effective edge set of slot $v^{\prime }$ as $K_{v^{\prime }}^{+}=\left\{k|U_{k,v}=1,k\in K^{+}\right\}$, then the decodable condition of overload slot $v^{\prime }$ after the *j*-th SIC is:(10)$$L_{v^{\prime }}^{j}-\left|K_{v^{\prime }}^{+}\right|\leq T_{th}.$$
If this condition is satisfied, peel the known interference off on this slot:(11)$$y_{v^{\prime }}^{\mathrm{SIC}}=y_{v^{\prime }}-\sum_{k\in K_{v^{\prime }}}x_{k}^{\mathrm{SIC}}.$$
Then send $y_{v^{\prime }}^{\mathrm{SIC}}$ to the intra-slot decoder in Section 3.2.1, where the reliable decoded message ${\hat{w}}_{v^{\prime },k}$ can be recovered. Consequently, the user node subsets $K_{j}^{+}$ and $K_{j}^{-}$ can be updated by the backward messages. Perform this process on each overload slot $v^{\prime }\in V^{-}$. As a result, $|K_{j}^{-}|$ will decrease after each iteration, for the underload slots are always helping the overload ones by message propagation on the factor graph. Under appropriate conditions, the SIC can converge to the required level.

## 4. Performance Analysis and Parameter Optimization

The goal of the coding parameter optimization problem of the proposed scheme is to minimize the SNR threshold under a given PUPE (in Section 2). However, due to the complicated intra/inter-slot encoding structure, the relationship between the coding parameters and the performance indicator is not explicit. Therefore, this section addresses the problem by error event decomposition [16]. By breaking the system-level PUPE down to module-level error rates, the error contribution of each module can be analyzed and correlated with their parameters and eventually form a system-level parameter optimization problem.

### 4.1. Error Rate Analysis by Decomposition

According to the decoder structure described in Section 3.2, the PUPE in (Equation 1) can be decomposed into four parts, as depicted in Figure 5.

Although, as described in Algorithm 1, the decoding structure consists of the compound intra/inter-slot iteration, it can nonetheless be regarded as a three-stage process when analyzing errors due to its successive style. The first stage is the CS pilot recovery, where the pilot resource collision $d_{k_{1}}=d_{k_{2}},k_{1}\neq k_{2}$ occurs at the transmitter side and the support recovery error ${\hat{d}}_{k}\neq d_{k}$ at the receiver side. The pilot plays three significant roles: the first part of the user payload, the user-specific IDMA interleaver, and the pointer used to reconstruct the packet-level factor graph. Thus, the pilot error will cause not only packet loss of its user but also the chain effect spreading to the correlated packets of other users. The IDMA decoding error ${\hat{b}}_{k}\neq b_{k}$ and the remaining error after the intra-slot SIC process ${\hat{b}}_{k}^{\mathrm{SIC}}\neq b_{k}$ are both conditioned on the successfully recovered pilots. Then the modular errors are expressed as the functions of their corresponding coding parameters in (Equation 12).
(12)$$\begin{array}{cc} P\left(w_{l}\neq {\hat{w}}_{l}\right) & =P\left\{d_{k_{1}}=d_{k_{2}}\right\}+P\left\{{\hat{d}}_{l}\neq d_{l}\right\}\\ & +P\left\{{\hat{b}}_{k}\neq b_{k}|{\hat{d}}_{l}=d_{l}\right\}+P\left\{{\hat{b}}_{k}^{\mathrm{SIC}}\neq b_{k}|{\hat{d}}_{k}=d_{k}\right\}\\ \; & =\epsilon_{1}\left(N_{s}\right)+\epsilon_{2}\left(N_{s},B_{s}\right)+\epsilon_{3}\left(R_{L},R_{r}\right)+\epsilon_{4}\left(\lambda \left(x\right)\right) \end{array}$$

The CS pilot parameters $N_{s}$ and $B_{s}$ are restricted by the collision and detection conditions. $\epsilon_{1}$ is the resource collision avoiding condition in [16]. The length of the separated pilot info part from user payload $B_{s}$ should be large enough to provide non-collision patterns. As for $\epsilon_{2}$, the dimension of the sensing matrix is bounded by the Restricted Isometry Property (RIP) [31] condition. In other words, when $T_{th}$ is fixed, large row dimension $N_{s}$ can reduce the $\epsilon_{2}$. Thus, we can conclude that $N_{s}\propto T_{th}$ and $B_{s}\propto T_{th}$.

The IDMA block error rate $\epsilon_{3}$ is the function $\epsilon_{3}=P\left(B_{c},R_{L},R_{r},L_{v},\mathrm{SNR}\right)$. According to the classical analysis of IDMA, $\epsilon_{3}$ is based on the single-user performance of the inner rate-$R_{L}$ FEC code and degrades with the gradually severe interference caused by the increase of the number of superimposed users $L_{v}$. The SNR cost of $\epsilon_{3}$ at a required level can be extracted on the performance curve of a given threshold $T_{th}$ by simulation.

### 4.2. Inter-Slot Degree Distribution Analysis

As the SNR cost of the IDMA system increases with $T_{th}$, the SIC decoding on the compound factor graph of the inter-slot code can further reduce the threshold $T_{th}$ to reach the same level of PUPE, resulting in the reduction of SNR threshold. Thus, degree distribution optimization aims to minimize $T_{th}$.

First, we start with an idealized asymptotic investigation. Previously the T-Fold IRSA was modeled as a factor graph, on which the user node degree distribution λ(x) determines the slot node degree distribution ρ(x):(13)$$\begin{array}{cc} \rho (x) & =\sum_{i=r}^{R_{\max}}\rho_{r}x^{r}=\sum_{r=1}^{R_{\max}}P\left(L_{v}=r\right)x^{r}\\ \rho_{r} & =P\left(\mathrm{Bino}\left(\lambda^{\prime }\left(1\right)K_{a},1/V\right)=r\right) \end{array}$$
where Bino(·) denotes the binomial distribution. The probability of high-degree slot nodes increases with the average packet repetition rate $\lambda^{\prime }\left(1\right)$ and decreases with more slots *V*, representing intensified superposition. Figure 6a gives an example of this rule.

Moreover, the convergence behavior of the SIC on the factor graph can be characterized by density evolution (DE), which is similar to the LDPC message passing procedure under the erasure channel [32]. At the *t*-th iteration, the erasure probability of the slot nodes is $\phi_{t}$ and $\eta_{t}$ for the user nodes:(14)$$\begin{array}{cc} \phi_{t}\left(\eta_{t}\right) & =1-\left(\sum_{r=1}^{T_{th}}\rho_{r}+\sum_{r>T}\rho_{r}\sum_{i=0}^{T_{th}-1}C_{r-1}^{i}{\left(1-\eta_{t}\right)}^{r-1-i}\eta_{t}^{i}\right)\\ \eta_{t+1}\left(\phi_{t}\right) & =\lambda \left(\phi_{t}\right). \end{array}$$
where $x_{0}=1$ at the beginning without a priori knowledge of user nodes, and ρ(x) can be derived from λ as in (Equation 13). Under appropriate λ(x), the erasure probability of slot nodes $\phi_{t}$ gradually descends with iteration. As $\lambda (\phi_{t})$ is a positive-coefficient polynomial function, $\eta_{t+1}\left(\phi_{t}\right)$ is monotonically increasing, indicating that $\eta_{t}$ also converges to 0 with descending $\phi_{t}$. Meanwhile, the higher the threshold $T_{th}$ is, the higher the decodable probability of slot nodes $\sum_{r=1}^{T_{th}}\rho_{r}$ is. Notice that the two-probability expression corresponds to the two message-passing procedures in Algorithm 1, respectively. Therefore, the SIC iteration can make as many overload user nodes decodable as possible. An example of the SIC procedure simulated by density evolution is depicted in Figure 6b. When *V* and λ(x) are fixed, the cost of carrying more users $K_{a}$ is the increase of threshold $T_{th}$ to guarantee convergence under limited iterations, i.e., the rise of IDMA’s SNR requirement.

The tool to determine the convergence condition of density evolution iteration is its EXIT (EXtrinsic Information Transfer) chart [33]. Under given $K_{a}$, *V* and $T_{th}$, plot the erasure probability curves $\phi^{-1}\left(\theta \right)$ and η(θ) in (Equation 14) on one chart, and find a trajectory between those two curves starting from η(1)=1. If the trajectory reaches η(0)=0, this λ(x) enables the SIC iteration to converge, conversely not. Figure 6c,d shows a boundary condition case where the iteration tunnel is about to close for $T_{th}$. Since $\lambda {\left(x\right)}^{\prime \prime }\left(x\right)>0$, η(θ) is always concave. Moreover, this effect becomes stronger when the weights of higher-order coefficients increase. However, it is not easy to explicitly express the boundary condition. We adopt the numerical method, differential evolution [34], to solve the implicit equation and search the range of suitable λ(x):(15)$$\begin{array}{cc} \lambda^{*}\left(x\right) & \in \{\lambda \left(x\right)|\phi^{-1}\left(\theta \right)-\eta \left(\theta \right)>0,\theta \in \left(0,1\right)\}\\ \;s.t.\;\; & \phi \left(\theta \right)=1-\left(\sum_{r=1}^{T}\rho_{r}+\sum_{r>T}\rho_{r}\sum_{i=0}^{T-1}C_{r-1}^{i}{\left(1-\theta \right)}^{r-1-i}\theta^{i}\right)\\ \; & \eta \left(\theta \right)=\lambda \left(\theta \right),\;\lambda_{i}\geq 0,\;T\in N^{+}\\ \; & \rho_{r}=P\left(\mathrm{Bino}\left(\lambda^{\prime }\left(1\right)K_{a},1/V\right)=r\right) \end{array}$$

Then we move further to practical consideration. An important preassumption for density evolution analysis is that $K_{a},V\to \infty$ ensures the isotropy of distribution, while in practice, they are limited. It is challenging for the user-independent random schedulers to guarantee $\beta_{k}∼\lambda \left(x\right)$ when $K_{a}$ is relatively small, especially for the repetition times with low probability (high-order degrees). On the other hand, the finite length effect causes short cycles and trapping loops in the randomly formed factor graph. In some worse cases, the SIC iteration cannot start or converge. Although the effect of $\epsilon_{3}$ can be ignored when considering boundary conditions, the remaining errors on the underload slot nodes may cause error propagation. Nonetheless, $\Lambda^{*}$ gives a basic scope, which just needs to be narrowed.

We use Monte-Carlo simulation to practically examine the effectiveness of candidates in $\Lambda^{*}$. Under finite $K_{a}$, *V*, and $\epsilon_{3}$, the post-iteration packet loss rate of user nodes obtained by simulation is the actual PUPE. Besides, the cost of the packet diversity gain is the additional energy spent on repeated packets. The SNR threshold after inter-slot encoding is:(16)$$E_{b}/N_{0}={\left(E_{b}/N_{0}\right)}_{T_{th}}\lambda^{\prime }\left(1\right)$$
where ${\left(E_{b}/N_{0}\right)}_{T_{th}}$ is the SNR when $\epsilon_{3}$ reaches an effective level under threshold $T_{th}$. After that, the trade-off of intra-slot encoding should also be considered.

### 4.3. Joint Parameter Optimization Algorithm

Integrating the analysis on the above modules, the complete parameter optimization procedure is proposed as Algorithm 2.
**Algorithm 2** Joint optimization of coding parameters**Require:** the legnth of user payload *B*, number of actove users $K_{a}$, frame length $N_{\mathrm{tot}}$, and target PUPE ϵ.**Ensure:** the length of CS pilot info part $B_{s}$, length of CS pilot $N_{s}$, LDPC code rate $R_{L}$, intra-slot repetition rate $R_{r}$, and inter-slot packet spreading distribution λ(x).1:Initialization: Determine $N_{s}$ and $B_{s}$ by the CS decoding conditions, randomly chose rate configuration $\left\{R_{L},R_{c}\right\}$, and calculate *V*;2:**while** 
$R>R_{\min}$ 
**do**3:   **for** $T_{th}$ from $T_{\min}$ to $T_{\max}$ **do**4:     Obtain $\epsilon_{3}$ and ${\left(E_{b}/N_{0}\right)}_{T_{th}}$ by IDMA simulation;5:     Search the convergable range $\Lambda^{*}$ using density evolution by (Equation 15);6:     **for** λ(x) from $\min\lambda^{\prime }\left(1\right)$ to $\max\lambda^{\prime }\left(1\right)$ **do**7:       Perform Monte-Carlo simulation to validate λ(x) under the residual error $\epsilon_{3}$;8:       Output the post-SIC-iteration error rate $\epsilon_{4}$;9:       **if** $\epsilon_{4}\leq \epsilon$ **then**10:        Reserve λ(x) and break;11:      **end if**12:    **end for**13:  **end for**14:  **if** SNR threshold (Equation 16) increases **then**15:    Choose the $\left\{R_{L},R_{c}\right\}$ pair with higher total rate *R*, adjust *V* accordingly;16:  **else**17:    Lower the total rate *R*, adjust *V* accordingly.18:  **end if**19:**end while**20:Calculate the SNR threshold by (Equation 16) and output the optimal coding parameters.

Overall, we use a heuristic bootstrap method to jointly optimize the coding parameters analyzed above. The CS pilot configurations control $\epsilon_{1}$ and $\epsilon_{2}$, basically determined by $K_{a}$. To tackle the contradiction between intra-slot coding gain and inter-slot diversity gain, the IDMA rate *R* increases with iterations, while the inter-slot diversity decreases in each iteration. The initial rate *R* is randomly chosen, and then *V* can be determined. ${\left(E_{b}/N_{0}\right)}_{T_{th}}$ at $\epsilon_{3}$ is extracted on simulation curves. The convergence condition is ensured by (Equation 15). Then the effectiveness of λ(x) with increasing energy cost is checked by simulation until it satisfies the post-SIC-iteration PUPE requirement. If the SNR threshold raises compared with the last iteration after optimization, *R* should be increased to enhance the inter-slot code. If not, lower *R* to reduce the multi-user interference.

## 5. Numerical Results

In this section, we evaluate the proposed scheme using numerical indexes of two aspects: the $K_{a}$-SNR threshold curve representing energy efficiency and the FLOPf (FLoating-point Operations Per frame) comparison representing computational complexity.

### 5.1. Energy Efficiency Analysis

The SNR threshold, by definition, is the minimum required SNR that achieves a target PUPE under specific configurations. As described in Section 2, the $K_{a}$-SNR curve is acquired under some fundamental constraints. Thus, we give the primary scenario configurations in Table 1.

These parameters are shared in the following simulations. To get the SNR threshold for each $K_{a}$, we optimize the coding parameters by Algorithm 2 point-by-point. The results are displayed in Table 2. Under all the configurations, $\epsilon_{1}<10^{-3}$ and $\epsilon <10^{-4}$, so that the pilot error would not affect the subsequent decoders. Through adjustment, the intra-slot coding gain and inter-slot diversity gain are balanced, the sum of which reaches the optimal point.

It can be observed that the proportion of the two types of gains varies with $K_{a}$. In the low $K_{a}$ region, the intra-slot coding gain is more effective against multi-user interference, and the cost of reducing *V* is affordable. However, when it comes to the high $K_{a}$ region, the main problem is to tackle the rise of threshold $T_{th}$ by the inter-slot SIC. Meanwhile, the severe superposition of packets requires *V* to increase, so the inter-slot diversity gain dominates.

Next, we compare the proposed scheme with several existing representatives in Figure 7a. Based on the T-Fold Aloha scheme, the CCS-AMP [36] uses CS as the intra-slot encoder to resolve superposition and enhance the AMP algorithm. As introduced in Section 1, the CCS-AMP scheme is based on the T-Fold Aloha scheme, while SC-LDPC + SIC [20] and Polar-IRSA [21] are T-Fold IRSA. Sparse IDMA [13] and Polar-sparse spreading (Polar-SS) [15] are in the random spreading paradigm. Polar-IRSA achieves the best performance in the IRSA category with a better intra-slot code design. The excellent performance of polar codes in short length makes polar code base schemes stand out in low $K_{a}$ conditions where the intra-slot coding gain is critical. Our scheme occupies the middle position among the three for its repetition diversity in intra-slot IDMA code provides more coding gain than pure LDPC. The random spread spectrum schemes have the best performance of all paradigms, which can be decomposed into a cascaded structure on one frame-length long time slot. The sparse outer code can eliminate multi-user interference and provide a particular coding gain in the low $K_{a}$ region.

As $K_{a}$ increases, our scheme maintains the lowest slope and eventually achieves a performance advantage in the high $K_{a}$ region. This is mainly because we effectively control the increase of the threshold $T_{th}$ by utilizing fine optimization of the intra-slot gain and the inter-slot gain. The outer code of the sparse spreading schemes tends to be rateless, and its sparsity and gain gradually diminish. Meanwhile, other IRSA schemes only exploiting packet-level diversity suffer the same problem, let alone the bit-level spreading outperforms the packet-level one.

### 5.2. Complexity Analysis

Based on the analytical framework proposed in [10], the FLOPf can represent the complexity of the decoder. For each encoding scheme, FLOPf can be expressed as a function of its coding parameters. We compare the proposed scheme with Polar-RS, Polar-IRSA, and CCS-eAMP, the expressions of which are listed in Appendix A. When comparing these schemes, it is insightful to investigate the complexity cost of the corresponding energy efficiency gain. Figure 7b shows the SNR level and the required FLOPf of each scheme when the number of users $K_{a}$ reaches 150.

Although our scheme’s and CCS-AMP’s performance are the same, the latter costs about ten times more FLOPf than the former. Meanwhile, sparse IDMA spends 100 times more complexity for 0.4dB energy gain, while for the polar-RS scheme, it is $10^{6}$ for 0.8 dB. The intra-slot decoder is a linear iteration and converges after 3 to 5 iterations due to the high coding gain. Besides, the inter-slot iteration is restricted to 10 times by the joint degree distribution optimization. Therefore, our scheme achieves a better energy-complexity trade-off.

## 6. Conclusions

The proposed coding scheme achieves performance gain in the high $K_{a}$ region, benefitting from intra/inter-slot gain balance under jointly optimized parameters. Although it is not as good as existing solutions in the low $K_{a}$ region, it achieves a better trade-off between performance and complexity. The joint design of intra/inter-slot code exploits the potential of the T-Fold IRSA scheme more thoroughly. Thus, our scheme would be a prospective candidate for UMA design in next-generation IoT.

For future research, we can consider the fading scenario. Our scheme can easily be promoted to the block fading channel because of two inherent advantages: 1. Inserting the pilot at each slot means the channel response of each user at each slot can be independently estimated; 2. The intra-slot IDMA is a universal code. After optimizing it under the AWGN channel, its configuration can be directly applied to the fading scenario.

## Figures and Tables

**Figure 1 sensors-23-00242-f001:**
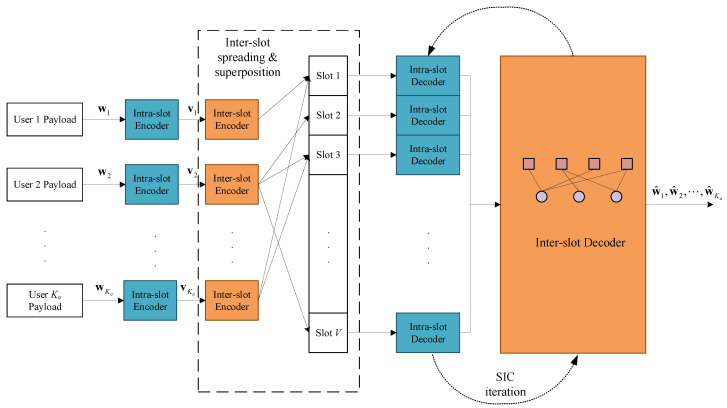
The overall system framework of the proposed scheme.

**Figure 2 sensors-23-00242-f002:**
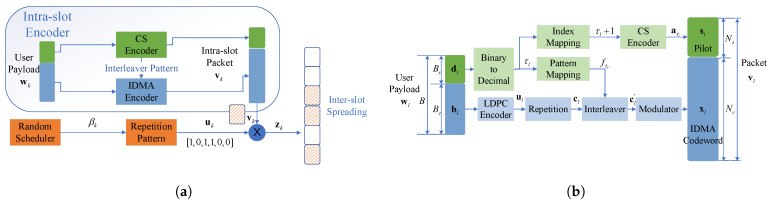
(**a**) The overall structure of the encoder. (**b**) The Intra-slot encoder.

**Figure 3 sensors-23-00242-f003:**
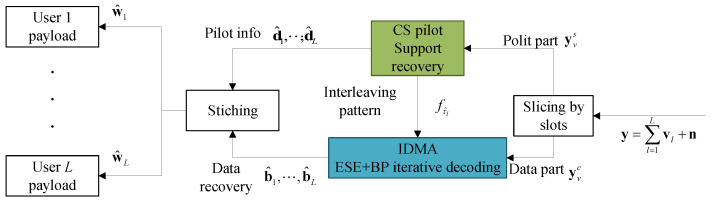
Intra-slot decoder structure at slot *v*.

**Figure 4 sensors-23-00242-f004:**
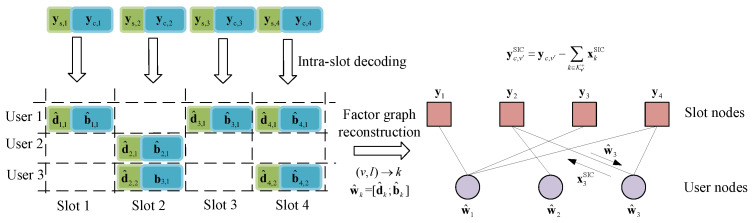
An example for inter-slot factor graph reconstruction and SIC process, where $K_{a}=3$ and V=4.

**Figure 5 sensors-23-00242-f005:**
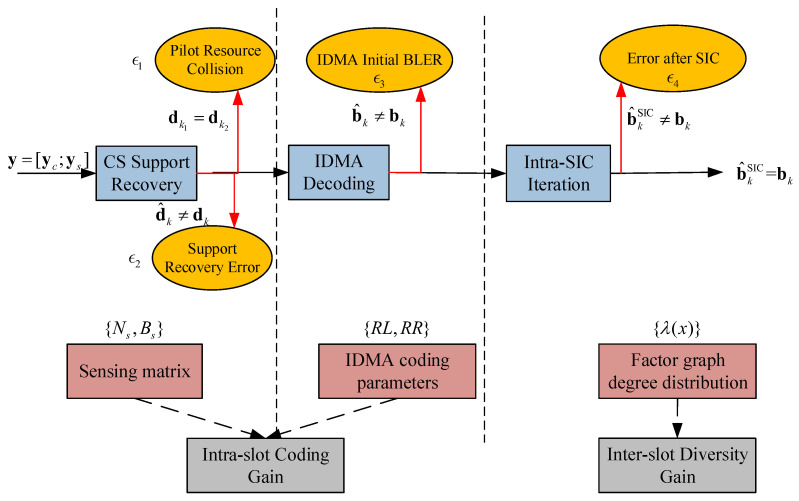
The system-level packet loss broken down into module-level error events, with the corresponding coding parameters and coding gains.

**Figure 6 sensors-23-00242-f006:**
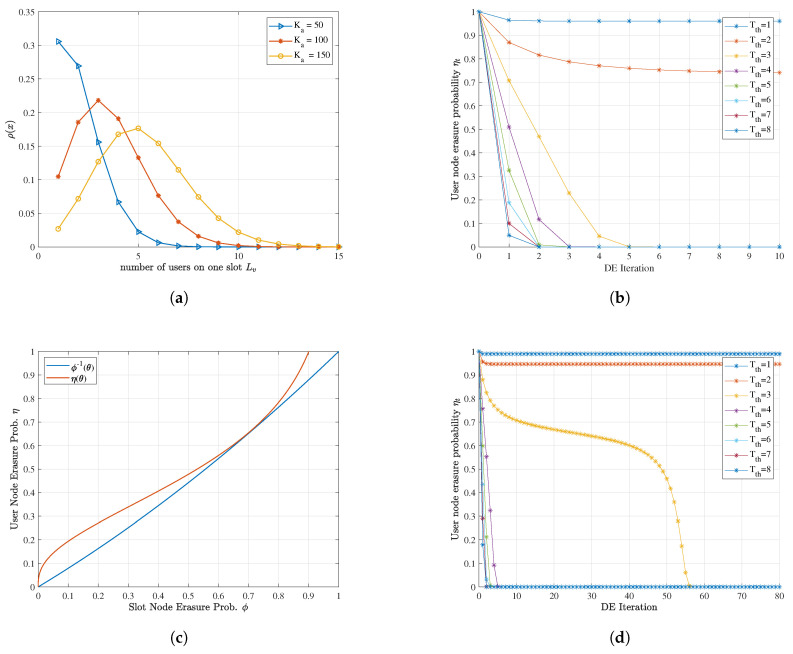
Density evolution analysis of inter-slot code: (**a**) The degree distribution of slot nodes ρ(x) at $\lambda \left(x\right)=0.5x^{2}+0.5x$ and different users. (**b**) The density evolution procedure at $\lambda \left(x\right)=0.5x^{2}+0.5x$, $K_{a}=100$ and V=28, when $T_{th}<3$ it does not converge. (**c**) The density evolution EXIT chart at $\lambda \left(x\right)=0.23x^{2}+0.77x$, $K_{a}=150$, V=28 and $T_{th}=3$. The tunnel is about to close, representing the boundary condition. (**d**) Corresponding density evolution iteration procedure at the same configuration with (**c**) but varying $T_{th}$, where $T_{th}=3$ is just able to converge.

**Figure 7 sensors-23-00242-f007:**
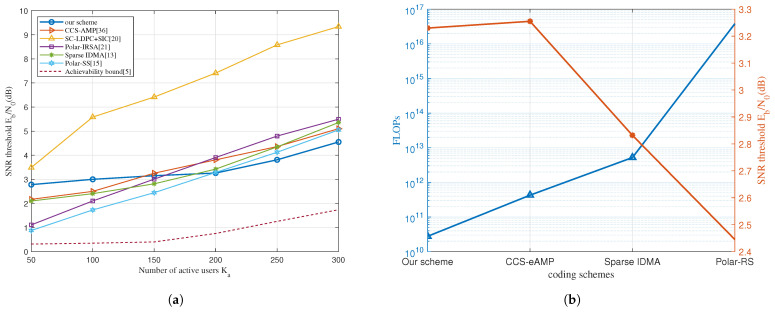
(**a**) Comparison of $K_{a}$-SNR threshold curves of different coding schemes. $N_{tot}=$ 30,000, B=100, PUPE = 0.1. (**b**) Performance and computational complexity comparison at $N_{tot}=$ 30,000, B=100, $K_{a}=150$, PUPE = 0.1.

**Table 1 sensors-23-00242-t001:** Primary scenario and configurations in numerical simulations.

Parameters	Configurations
Total frame length $N_{tot}$	30,000
User payload length *B*	100
LDPC encoder	5G NR BG2 [35]
Number of active users $K_{a}$	50 to 300
IDMA modulation	BPSK
CS pilot sensing matrix	Gaussian random matrix
PUPE requirement ϵ	0.1

**Table 2 sensors-23-00242-t002:** Optimized coding parameters of the proposed scheme.

Parameters	Optimized Results					
**Number of active users** $K_{a}$	**50**	**100**	**150**	**200**	**250**	**300**
Pilot info length $B_{s}$	11	11	11	11	12	12
IDMA data info length $B_{c}$	89	89	89	89	88	88
CS pilot length $N_{s}$	100	100	100	100	125	125
LDPC rate $R_{L}$	1/5	1/5	1/3	1/3	2/5	1/4
Intra-slot repetition rate $R_{r}$	1/2					
IDMA rate *R*	0.1	0.1	0.167	0.167	0.2	0.25
Number of slots *V*	28	28	43	43	48	57
threshold $T_{th}$	2	3	3	3	4	5
Optimized degree distribution	$0.11x^{2}+0.89x$	$0.12x^{2}+0.88x$	$0.13x^{2}+0.87x$	$0.15x^{2}+0.85x$	$0.18x^{2}+0.82x$	$0.2x^{2}+0.8x$
Average packet repetition rate	1.11	1.12	1.13	1.15	1.18	1.20
SNR threshold	2.75	2.89	3.23	3.91	4.11	4.79

## Data Availability

Not applicable.

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
