# Peer review of "Joint Intra/Inter-Slot Code Design for Unsourced Multiple Access in 6G Internet of Things"

_sensors, 2022, doi:10.3390/s23010242_

Round 1

Reviewer 1 Report

This work designs an intra-slot/inter-slot scheme and optimizes the parameters of this joint scheme. The paper is well written. Following are some minor comments.

1. The problem to be solved in this paper is not clear. Please specify it. It would be better if the author adds some diagrams to better see what is the problem is and what your solution.

2. Though the method is introduced clearly, the comparison with other related work, especially the coding and decoding used in the UMA, is not given. Then, the contribution and effectiveness cannot be well evaluated.

3. The characters in figures are too small, especially Figures 1-2.

4. The expression of equations is not canonical, such as Eq. (13).

Author Response

Thank you for your comments on this article, and here are the point-by-point responses to your comments:

  1. The problem to be solved in this paper is not clear. Please specify it. It would be better if the author adds some diagrams to better see what is the problem is and what your solution

Reply: The problem to be solved in this paper is to propose a novel UMA (Unsourced Multiple Access) coding schemes to achieve better energy efficiency and lower complexity than existing solutions. As we emphasized in the Abstract and Introduction, the multi-user coding scheme design is a critical problem for UMA.

Under this target, we tackle this problem with three aspects of designing improvement under the T-Fold IRSA paradigm (as detailed description in section 1.3). Aspects 1 and 2 are reflected in the design of coding and decoding respectively, and mainly correspond to Section 3, while aspect 3 is reflected in the method of coding parameters optimization, mainly corresponding to Section 4.

The energy efficiency index is defined by the SNR threshold corresponding to the number of users, as described in Section 2. This index is bounded by the achievability bound proposed by [5], as the limit of any UMA system. Thus, we put the proposed scheme together with related solutions under this analysis framework, to verify and compare their energy efficiency features. As for complexity, we propose the FLOPf index and show the complexity indexes of different schemes to achieve the corresponding energy efficiency.

  1. Though the method is introduced clearly, the comparison with other related work, especially the coding and decoding used in the UMA, is not given. Then, the contribution and effectiveness cannot be well evaluated.

Reply: In this paper, the comparisons with other related works are given in two aspects, methodology (in Section 1 Introduction) and the tests of effects (in Section 5 Numerical Results). As we did in Section 1.2, the existing UMA coding schemes can be categorized into three paradigms. The differences between paradigms are also compared there. When comparing across paradigms, these conclusions and analyses are the main determining factors. Within the T-Fold IRSA paradigm, we emphasized our contributions in three aspects, as mentioned in Section 1.3.

It should be noted that since our approach involves systematically designing the co-encoding and joint-decoding structure, and the final effect is reflected in the systematic index, the comparison between coding schemes can only be systematic. For example, we cannot simply compare the packet loss rate between IDMA and SC-LDPC to prove the superiority of which scheme. These systematic indicators (energy efficiency and complexity) are compared with existing encoding and decoding schemes under a unified framework, as mentioned in the previous reply.

  1. The characters in the figures are too small, especially Figures 1-2.

Reply: The scales of Figures 1-2 are adjusted for a better reading experience.

  1. The expression of equations is not canonical, such as Eq. (13).

Reply: The typo in Eq. (13) is fixed.

Reviewer 2 Report

1. Acronyms should be defined before citing them

2. Please define ALOHA

3. Figure 2 not mention in the text

4. More information should be provided about Table 1

5. The values quoted in Table 2 are they arbitrary values? Please justify

6. Additional complexity due to interference and energy collapse are not incorporated in the design. Please clarify?

Author Response

Thank you for your comments on this article, and here are the point-by-point responses to your comments:

  1. Acronyms should be defined before citing them
  2. Please define ALOHA

Reply for 1&2: ALOHA is a common random-access protocol whose name comes from the English word aloha. In some literature, ALOHA is written in all caps to indicate its use as a communication protocol. "ALOHA" is used in the article, but it seems to have created an ambiguity that makes it seem like an abbreviation of some sort. Therefore, we change ALOHA to Aloha in the whole article and use it as a word to avoid ambiguity. In addition, as for ALOHA protocol, reference [11] is also used for its first appearance in this paper, which has detailed knowledge of all types of ALOHA protocols used in this paper.

  1. Figure 2 not mention in the text

Reply: Description has been added at the corresponding place in the text.

  1. More information should be provided about Table 1

Reply: The configured parameters in Table 1 correspond to the coding design parameters in the previous chapter. What details do you think need to be added? Or a specific description of the parameters?

  1. The values quoted in Table 2 are they arbitrary values? Please justify

Reply: As described in lines 342 to 344, for each Ka, these values are obtained by the optimization algorithm proposed in Section 4. The caption of the table is modified.

  1. Additional complexity due to interference and energy collapse is not incorporated in the design. Please clarify?

Reply: The complexity index is determined by the receiving algorithm itself. As shown in the appendix, the FLOPf of each encoding scheme can be expressed as an expression composed of its corresponding encoding parameters, and these expressions can be directly derived from the decoding algorithm. Therefore, the way to handle multi-user interference and energy collapse is included in the design of the corresponding encoding and decoding algorithm, rather than an “additional” factor.

Reviewer 3 Report

It is a well-presented paper. I want to recommend it for publication.

The author presented Unsourced multiple access (UMC) scheme for Intra and inter-slot coding. This scheme helps in improving SNR and throughput of the communication in Aloha. The proposed system is tested using analytical and simulation models. the Results are also very satisfactory.

The overall quality of the paper is good and can be accepted in its current form. 

Author Response

Thank you for your comment.